# Potential Adverse Drug Events Identified with Decision Support Algorithms from Janusmed Risk Profile—A Retrospective Population-Based Study in a Swedish Region

**DOI:** 10.3390/pharmacy12060168

**Published:** 2024-11-15

**Authors:** Tora Hammar, Emma Jonsén, Olof Björneld, Ylva Askfors, Marine L. Andersson, Alisa Lincke

**Affiliations:** 1The eHealth Institute, Department of Medicine and Optometry, Linnaeus University, S-391 82 Kalmar, Swedenolle.bjorneld@regionkalmar.se (O.B.);; 2Linnaeus University Centre for Data Intensive Sciences and Applications (LnuC DISA), Department of Computer Science and Media Technology (CM), Faculty of Technology, Linnaeus University, S-391 82 Kalmar, Sweden; alisa.lincke@lnu.se; 3Business Intelligence, IT Division, Region Kalmar County, S-392 32 Kalmar, Sweden; 4Division of Clinical Pharmacology, Department of Laboratory Medicine, Karolinska Institute and Clinical Pharmacology, Medical Diagnostics Karolinska, Karolinska University Hospital, S-141 86 Stockholm, Sweden; marine.andersson@regionstockholm.se

**Keywords:** adverse drug events, clinical decision support system, drug-related problems, pharmacoepidemiology, side effects, drug–drug interactions

## Abstract

Adverse drug events (ADEs) occur frequently and are a common cause of suffering, hospitalizations, or death, and can be caused by harmful combinations of medications. One method used to prevent ADEs is by using *clinical decision support systems* (CDSSs). Janusmed Risk Profile is a CDSS evaluating the risk for nine common or serious ADEs resulting from combined pharmacodynamic effects. The aim of this study was to examine the prevalence of potential ADEs identified using CDSS algorithms from Janusmed Risk Profile. This retrospective, cross-sectional study covered the population of a Swedish region (*n* = 246,010 inhabitants in year 2020) using data on all medications dispensed and administered. More than 20% of patients had an increased risk of bleeding, constipation, orthostatism, or renal toxicity based on their medications. The proportion of patients with an increased risk varied from 3.5% to almost 30% across the nine categories of ADEs. A higher age was associated with an increased risk of potential ADEs and there were gender differences. A cluster analysis identified groups of patients with an increased risk for several categories of ADEs. This study shows that combinations of medications that could increase the risk of ADEs are common. Future studies should examine how this correlates with observed ADEs.

## 1. Introduction

Medications are an important part of health care and improve the lives of many patients. However, adverse drug events (ADEs) occur frequently and are a common cause of suffering, hospitalizations, and death [1,2,3,4]. ADEs are defined as harm caused by a drug or the inappropriate use of a drug [5], including adverse drug reactions (ADRs) and pharmacokinetic drug–drug interactions (DDIs) and pharmacodynamic DDIs, among other things [6]. Pharmacokinetic interactions are when medications affect each other via absorption, distribution, or elimination, and pharmacodynamic interactions are when, e.g., drugs have additive or synergistic pharmacological effects, causing similar effects or side effects. The prevalence of DDIs among hospitalized patients is, according to a recent meta-analysis, on average, 65%, and the prevalence of ADEs caused by DDIs is around 17% [7], but results vary between studies [8,9]. A systematic review estimated that 19% of patients experience ADEs during hospitalization [4].

Thus, some of the ADEs can be the result of a single medication, while others are a result of the combination of two or more medications. Many ADEs may be prevented if we could identify and thus avoid potentially harmful combinations of medicines or inappropriate dosing [10,11]. One method to prevent harmful combinations of medications is by using *clinical decision support systems* (CDSSs) in health care or at pharmacies that can detect potential ADEs [12,13,14,15]. Sweden is at the international forefront in developing high-quality knowledge databases for clinical decision support, with most of them being a part of *Janusmed knowledge* databases [16,17,18]. However, they are underused tools and most Swedish regions only have a few of them integrated in their *electronic health records* (EHRs). There are many reasons why CDSSs are not used optimally, such as incompatibility with other systems, interruption in workflow, and overalerting by, e.g., giving too many non-relevant warnings, which might, in turn, lead to alert fatigue [19,20,21,22,23]. There are different strategies used to increase alert acceptance [21,24]. One way to improve the relevance of signaling by the CDSS is by studying occurrence and clinical outcomes in a large patient population. Preferably, the clinical and economical benefit of the CDSS should be documented to motivate the time spent on using the CDSS.

Janusmed Risk Profile is a rather new (2018) knowledge database developed to address this challenge by evaluating the risk for a number of common and/or serious side effects resulting from combined pharmacodynamic effects from medications used by the patient (i.e., pharmacodynamic interactions) [25,26]. In this paper, we refer to these as potential ADEs. The knowledge database gives risk classifications based on a patient’s list of current medications for nine pharmacological risk categories: *Anticholinergic side effects, Risk of bleeding, Constipation, Orthostatism, QT prolongation/arrhythmia, Nephrotoxicity, Sedation, Seizures*, and *Serotonergic side effects.* Each substance in the database has a risk score for each of the nine pharmacological risk categories, which are summarized according to an algorithm for a patient to provide a risk score for each category. Together, the risk levels for each of the nine categories are called the risk profile of the patient (Figure 1).

CDSSs have been shown to decrease ADEs in many clinical settings [12]. Janusmed Risk Profile has been shown to have a high specificity but low sensitivity when investigating clinical symptoms of sedation, constipation, orthostatic symptoms, and anticholinergic and serotonergic effects among elderly patients [26]. A previous study showed a decrease of 63% in the usage of drug combinations, with a risk for QT prolongation among elderly after the introduction of Janusmed Risk Profile into the EHR system [27]. However, a causal relationship between the introduction and the decrease could not be stated.

ADEs can be studied both as potential and observed (actual) [28,29]. Studies of potential ADEs give an indication of patients with an increased risk of ADEs, and can be based on patients’ current medications, sometimes referred to as potentially inappropriate prescriptions (PIPs) [30,31]. Observed (actual) ADEs can be studied based on clinical manifestations or documented events, either in clinical practice or using retrospective data [32]. There is a need to monitor the population for both potential and clinically observed ADEs when medications are used together, and to study which potential and actual ADEs are detected by CDSS algorithms. As the use of multiple medications, often referred to as polypharmacy [33,34], is increasing, it is important to enhance knowledge about the effects of combining these medications and to improve CDSSs to prevent ADEs.

The aim of this study was to examine the prevalence of potential ADEs, i.e., medications with an additive or synergistic pharmacological effect that could potentially increase the risk of common or serious side effects, identified using decision support algorithms from Janusmed Risk Profile. More specifically, this study aimed at answering the following research questions:

What proportion of the total population has received medications or combinations of medications that can increase the risk of ADEs according to Janusmed Risk Profile classifications?Are there differences in potential risks identified with Janusmed Risk Profile linked to gender and age?Are there groups of patients with an increased risk of many different ADEs?What medications are most involved in the alerts for potential ADEs?

## 2. Materials and Methods

### 2.1. Study Design and Population

This study was a retrospective cross-sectional study covering the population of the region of Kalmar County (hereafter referred to as Kalmar) at one point in time using data on prescription medications dispensed at pharmacies as well as administered within health care. Kalmar is one of 21 Swedish regions, located in the southern of Sweden, and had 246,010 inhabitants in the year 2020. For this study, the population was inhabitants in the region during a period of 120 days (8 July 2020–5 November 2020) that had at least one medication event (prescribed, dispensed, or administered medication) during that defined period (a total of 123,255 individuals, Figure 2). This defined period is hereafter referred to as *the 120-day period*.

For each individual, we combined data from the EHR on medications administered within health care and data on dispensed medications for the defined time period (including multi-dose drug dispensing) to obtain a complete list of current medications for each individual at the cut-off day. In Sweden, medications are often dispensed for 3-month use each time, but since there can sometimes be a little longer between occasions for dispensing, 120 days was selected as the time period for capturing concurrent use of medications. November 5 was regarded as the cut-off day and 120 days prior to that was used to identify patients’ current medications at one point. In Kalmar, the same EHR has been used in both primary health care and hospitals for more than a decade and has high-quality data available for research. Data on medications as well as patient information were extracted from the EHR. For dispensed medications, data were extracted from the eHealth Agency through the data warehouse in Kalmar. The data on dispensed medications from the eHealth Agency are already included in the data warehouse in Kalmar and updated on a monthly basis, and can therefore be linked on an individual level before data are delivered to the researchers. These data are almost the same as the data in the Swedish prescribed drug register, which are more commonly used in research [35]. These data include all prescription medications dispensed at any pharmacy in Sweden for inhabitants in Kalmar. They do not include over-the-counter (OTC) medications. Data on medications administered within health care (without a prescription being issued to a pharmacy) are not included in the ordinary dispensing data; they are instead documented in the EHR and need to be collected from the EHR database separately. Medications dispensed at pharmacies and administered within health care will, in this paper, be referred to as medication and medication events. For an overview of the different types of medication events and the data that we use regarding them, see Appendix A. Data from private health care providers and veterinary medications were excluded, as well as patients deceased before or during the 120-day period. For more details on the datasets, see Figure 2.

The Anatomical Therapeutic Chemical (ATC) classification system was used in this study. In the ATC classification, the active substances are classified into groups at five different levels according to the organ or system on which they act and their therapeutic, pharmacological, and chemical properties. In this study, the ATC fifth level was used.

### 2.2. Janusmed Risk Profile

The knowledge database and rule-based algorithms from Janusmed Risk Profile (previously known as Pharao) were applied to retrospective data for the defined population and 120-day period (Figure 2). For each individual, a list of concurrent medications at the cut-off day was created and analyzed with the algorithms of the Janusmed Risk Profile knowledge database. From that, we obtained data where each of a patients’ medications obtains a risk value (based on the substance) for each of the nine pharmacological risk categories: *Anticholinergic side effect, Risk of bleeding, Constipation, Orthostatism, QT prolongation, Nephrotoxicity, Sedation, Seizures*, and *Serotonergic side effect.* These are summarized according to an algorithm for each individual patient to provide a risk level for each of the nine categories, where level 0 is *no increased risk*, level I is *somewhat increased risk*, level II is *moderate increased risk*, and level III is a *significant increased risk* (the highest risk level). As a part of that, topical medications (based on drug form) were excluded, and only unique medications were included in the calculation toward risk, i.e., if the patient has received the same substance several times during the 120-day period, it will only be counted once. A unique substance in Janusmed Risk Profile is based on the top parent substance ID, which is similar to fifth-level ATC, but there can be differences. Together, the risk levels for each of the nine categories are called the risk profile of the patient. This decision support is not implemented in the EHR in Kalmar; it is only used in this study.

The Janusmed risk score calculations were based on rules that were documented in XML-files received from Janusmed Risk Profile management. Since new drugs are continuously introduced and new knowledge is discovered, Janusmed Risk Profile and its rules are not static and it has been released in several versions. We used the Janusmed Risk profile rules and algorithms from 10 January 2022, and the latest released version at that point (V3). The XML-files were loaded into corresponding database tables in the MS SQL database. By combining the loaded tables, we calculated the risk score for each medication event. All medication events for each patient were summarized in the 120-day period and the corresponding risk profile level for the cut-off day (5 November 2020) was calculated. The risk levels used in this study are only based on the Janusmed Risk Profile classifications and algorithms. Although these risk levels are believed to correspond to an increased risk of ADEs based on research and expert judgement when building the database, this study is not using data on actual or observed ADEs. Thus, this study is a descriptive study of potential ADEs.

### 2.3. Analysis

The medication dataset was extracted and analyzed after preprocessing in a Jupyter Notebook. Standard Python 3 libraries for data analysis, such as pandas and numpy, were used for this purpose. Data preprocessing involved the creation of a unique medication list for each patient, which was then utilized to compute the Janusmed Risk Profile (with nine risk categories).

The proportion with potential ADEs was calculated as the proportion of individuals in the region with at least one medication (according to definition in 2.1) event that, according to Janusmed Risk Profile, had an increased risk (risk level I, II, or III) for each of the nine categories of common or serious side effects. Descriptive statistics were summarized for the whole population and divided into age groups and gender. To examine if the difference between age groups and gender was statistically significant, two different non-parametric tests were used, due to the data not being normally distributed. For differences between males and females, the Mann–Whitney U-test was used, and, for differences between the three age groups, the Kruskal–Wallis test was used. This was performed for each of the nine risk categories, and the dependent variable was the risk level (0, 1, 2, 3).

To describe which medications were involved in the potential ADEs according to Janusmed Risk Profile, the top 10 substances (ATC 5th level) were calculated for each of the nine risk categories. The top ten medications involved in the increased risk were calculated as follows for each of the nine risk categories: only patients with risk level III (significant increased risk, i.e., the highest risk level) were selected. For each substance (ATC), the summarized risk value (i.e., the risk value in Janusmed Risk Profile for that substance * number of patients with that medication) was calculated and a list of the substances with the highest summarized risk value was compiled.

Clustering is an unsupervised machine learning technique for grouping objects with similar characteristics, and can also be used for structuring the unstructured data. In this study, we applied the k-modes clustering [36] approach to find groups of patients with similar medication risks. This technique defines clusters/groups based on the number of matching categories. The input for this model was a categorical vector of 9 values, which is the Janusmed Risk Profile, where each value is between 0 and 3 (0 indicates no risk, and 3 is a high risk). The number of clusters was defined using the elbow method [37], which helps to define the number of clusters with small variance. The descriptive analysis was performed to describe each obtained cluster. A clustering model was implemented in Python 3 using the *kModes* library [38] and *circlify* library for visualization of the clusters using the circle packing plot.

### 2.4. Analysis with Alternative Measure of Medications

Although the primary measure of simultaneous medications utilized in this study was a combination of medications dispensed at pharmacies and administered within health care during the defined 120-day period, analysis was made with two alternative ways of measuring: (A) only using data on dispensed medications at pharmacies for 120 days and (B) using data only on current medications in the EHR on the selected day of 5 November 2020, i.e., medication orders in the EHR medication list, that were active on that day (for a definition, see Appendix A). For all measures, Janusmed Risk Profile assessments were made in the same way.

### 2.5. Ethical Considerations

The research was approved by the Ethical Review Authority (no: 2021/03880) and data were handled accordingly to prevent unauthorized access and to minimize risk of individuals being identified. Informed consent of participants was not required as the retrospective study did not affect the health care of the included patients. Social security numbers (Swedish personal numbers) were not exposed during the work; instead, a pseudonymized code representing each unique patient was used.

## 3. Results

A total of 123,255 patients in Kalmar had at least one medication dispensed at a pharmacy or administered within health care during the defined 120-day period. The proportion with an increased risk of ADEs according to Janusmed Risk Profile varied between the nine risk categories, from 3.5% for *Serotonergic effect* to 28.6% for *Risk of bleeding* (Figure 3). The proportion of patients with the highest risk level (risk level III, significant increased risk) varied from 0.2% for *Renal Toxicity* to 6.8% for *Constipation*.

The proportion of patients with an increased risk (any level) was related to gender and age. For example, 33.9% of men were classified as having an increased *Risk of bleeding* compared to 24.5% of women (Figure 4). Similarly, the proportion with an increased *Risk of bleeding* ranged from 18.5% for patients younger than 65 to over half of patients aged 75 years or older (Figure 5). In contrast, the proportion with an increased risk of *QT prolongation* did not have such an obvious difference between age groups or between men and women. Statistical analysis of differences between gender showed that the difference was statistically significant for all nine risk categories (*p*-value for constipation = 0.001; for the other eight risk categories, *p*-value < 0.001). For the three age groups, there was a significant difference for all nine risk categories (*p* < 0.001).

### 3.1. Most Common Medications Among Patients with Highest Risk Level

To understand which medications were involved most in the increased risk for each of the nine categories, the 10 substances with the highest summarized risk value for patients with risk level III are shown in Table 1. The risk value for one substance can vary between 0 and 3, and the summarized risk value for a patient was used to show which medications contributed most, rather than which medications were most common in those patients. The results show that some medications contributing to risk level III have a high risk value on their own, and others have a low risk level but are instead more common medications.

Medications that contributed the most to the risk category *anticholinergic effect* included primarily psychotropic drugs such as antidepressants, anxiolytics, and antihistamines for systemic use, and an opioid (Table 1). For the category *Risk of bleeding*, medications contributing the most included antithrombotic agents and NSAIDs. Opioids and other medications affecting the opioid receptor are in the top of medications contributing to the risk category *Constipation*. Medications involved mostly in the risk category *Orthostatism* included mostly cardiovascular drugs like nitrates and agents acting on the renin-angiotensin system, but also other medications. For increased *QT prolongation*, there were many different medication groups contributing to the risk value in the category, including antiemetics, antibiotics, opioids, anxiolytics, and antidepressants. Among the medications contributing the most to potential *Renal toxicity* were proton pump inhibitors and NSAIDs. For the risk category *Sedation*, medications contributing the most included anxiolytics, sedatives, antidepressants, and opioids. Medications contributing the most to increased *Risk of seizures* were antipsychotics and immunosuppressants. Finally, for the risk category *Serotonergic effect*, the medications that contributed the most to the risk category were antidepressants, but also an antiemetic drug and an opioid. Some substances contributed to several of the risk categories; for example, oxycodone was among the top 10 medications for seven of the nine categories (Table 1).

### 3.2. Clustering Results

To visualize the total risk profile for each patient, a cluster analysis based on the risk levels for each of the categories was performed. Instead of just looking at one risk category at a time, this gave an overview of all the different risk categories of each patient. Ten clusters, each containing patients with a similar risk profile were identified using unsupervised machine learning (Figure 6). Table 2 describes characteristics for individuals in each cluster. For information about the five most common medications among patients in each of the 10 clusters, see the table in Appendix B. The largest cluster (cluster 0) contains primarily individuals with no increased risk for any of the risk profiles. In cluster 0, 56.5% of the patients are female, 18.3% are 75 years or older, they have a mean of 2.7 medications (unique ATC), and the mean sum of all their risk levels are 0.7. In contrast, cluster 2, the cluster with the highest proportion of patients aged 75 years or older (39.1%), contains many individuals with an increased risk of several of the risk categories. Patients in cluster 2 have a mean of 11.9 medications, and the mean sum of their risk levels for the nine categories is 11.5. The cluster with the highest sum of risk profiles is cluster 7. Although the patients in that cluster are younger than those in cluster 2, there is a higher proportion of patients with increased risk for several of the categories. Paracetamol was among the five most common medications in all clusters, atorvastatin in six of them, and omeprazole in five of the clusters (Appendix B).

### 3.3. Different Measures of Concomitant Use of Medications

The proportion of individuals assessed as having an increased risk of ADEs according to Janusmed Risk Profile differs depending on how simultaneous medications are measured (Appendix C). Only using data on dispensed prescriptions, which is a common method in pharmacoepidemiology, will identify fewer patients with potential ADEs. Using current medication orders found in the EHR on a specific day provides similar results to those a doctor would obtain if the decision support was integrated into the EHR and gave alerts based on the medication list (see Appendix A for the definition of the medication order). All prescriptions dispensed at a pharmacy and administered within health care require a medication order in the EHR.

## 4. Discussion

This study applied the rule-based algorithm of the knowledge database Janusmed Risk Profile to retrospective data on medications in Kalmar and showed that potential ADEs in the form of nine types of side effects are common. Among patients in Kalmar with at least one medication, in the selected 120-day period, more than 20% of patients had an increased risk of bleeding, constipation, orthostatism, or renal toxicity based on their medications, and more than 10% had an increased risk of anticholinergic effects, QT prolongation, and sedation. There were differences in the proportion, with an increased risk between age groups and gender. Results from this study show which medications contribute the most to patients obtaining the highest risk level and that some substances are common in several risk categories. A cluster analysis of patients’ risk profiles showed that there are groups of patients with an increased risk of several of the nine categories of side effects and that these are not only the oldest patients, although they have many medications.

A previous study investigated the risks according to Janusmed Risk Profile among the elderly in the region of Stockholm [27]. The same pattern of risks was found in that study as in the population in this current study. The frequency of increased risk was higher among the elderly, as expected. Another study in elderly patients in Finland compared the risk scores among the elderly with and without diabetes, and the pattern of risks found differed somewhat from this study. In that study, bleeding was most common, followed by constipation, anticholinergic effect, and orthostatism [39]. In the present study, we cover all groups of patients, which might explain why results differ. Previous studies show varying results regarding ADEs and gender differences, where some show an association between female gender and an increased occurrence of observed ADEs as well as prevalence of DDIs [4,11,40,41], while others show no difference [14]. In the present study, some of the risk categories of Janusmed were more common among men, and others among women, based on their medications. Conflicting results regarding gender differences might be explained by what type of ADEs are studied and how they are measured.

In the present study, we found a low proportion with the highest risk level (risk level III, significant increased risk) for renal toxicity (0.2%), a finding that is expected since co-treatment with at least three drugs with an increased risk is needed and there are only 165 substances with a risk value for renal toxicity.

The cluster analysis in the present study identified groups of patients with many medications and an increased risk for many types of ADEs according to Janusmed classification. Evidence for the association between polypharmacy and ADEs is conflicting [33,42]. It is important to study these groups of patients further to examine clinical outcomes indicating actual ADEs.

The proportion of the population classified as having an increased risk in any category provides an indication of how many alerts users (e.g., prescribers) would receive if Janusmed Risk Profile was implemented in clinical practice, such as in the EHR. This level of warnings can lead to alert fatigue, which may cause alerts to be ignored and ADEs to go unprevented, contrary to the intended purpose of the CDSS [15,24]. There are various approaches to improving the clinical relevance of the CDSS and thus reducing the risk of alert fatigue [21,43]. Results from population-based studies like the present study can offer some indication. However, to further validate and enhance the knowledge database, population-based real-world data should be used to correlate alerts (potential ADEs) with clinical outcomes (actual ADEs). Additionally, machine learning can be employed in various ways to reduce alert fatigue [44].

The strength with this study is the inclusion of the total population in the region at one point in time, using data that cover all dispensed medications as well as those administered in health care. This study also has several weaknesses, where many are common in pharmacoepidemiological studies. First, we do not know for sure that the medications referred to as simultaneous medications during the 120-day time period were actually used at that point in time, as some of the medications might have been used for a shorter time. Due to known adherence problems, it is likely that some of the medications were not being used. In addition, we do not have information about OTC medications. Using the active medication orders in the EHR instead would theoretically be a good way of capturing current medications on a specific day (so as to mimic the medication list of a patient on the selected day). However, in unpublished results, we see that a significant proportion of medication orders in the EHR should, in reality, be regarded as non-current medications as many of them will not be executed in the form of administrations, prescriptions, and dispensed prescriptions. This is also in line with other studies highlighting problems with errors in medication lists [45,46,47]. This study is based on data from Kalmar, one of the 21 regions in Sweden, at a single point in time. Although the results can provide an indication of the prevalence of potential ADEs, they cannot be fully generalized to all of Sweden, other countries, or different points in time. Another weakness in the present study is that it does not include drug dosage information or other patient characteristics in classifying risk. However, Janusmed Risk Profile does not take any of those factors into consideration in risk classification, and the aim of this study was to describe prevalence based on the current CDSS algorithms.

It is important to highlight that the present study only gives an overview of potential ADEs based on medication use, not actual events. Based on previous research [7,14,48], it is likely that only a small proportion of the patients end up having an actual ADE. Our research group is currently studying correlation between Janusmed risk classifications and actual observed clinical outcomes, and what factors, such as patient characteristics and co-morbidities, are associated with an increased risk of clinically manifested ADEs. The research group behind the present study is also exploring the potential in using machine learning models to improve predictions of ADEs. Furthermore, if (or when) Janusmed Risk Profile is implemented in Kalmar or any other region, a similar study should compare prevalence before and after implementation, as well as examine any effects on actual ADEs.

## 5. Conclusions

This retrospective, population-based study using CDSS algorithms from Janusmed Risk Profile demonstrated that combinations of medications that could increase the risk of ADEs are common. The proportion of patients with an increased risk varied from 3.5% to almost 30% across the nine categories of side effects (potential ADEs). More than 20% of patients had an increased risk of bleeding, constipation, orthostatic hypotension, or renal toxicity based on their medications. Age and gender were associated with the proportion having an increased risk. There are groups of patients at an increased risk for several of the risk categories, and these are not limited to the oldest patients, despite their extensive medication use. Some substances are common across multiple risk categories. The high prevalence of potential ADEs detected by the CDSS algorithm suggests that alert fatigue could be an issue when implemented. Future studies should examine the clinical relevance of the CDSS classifications, particularly their association with clinical outcomes indicative of actual ADEs. Additionally, future research should explore how predictions can be improved to make CDSSs more specific and reduce alert fatigue. The authors are currently addressing both of these aspects in ongoing research.

## Figures and Tables

**Figure 1 pharmacy-12-00168-f001:**
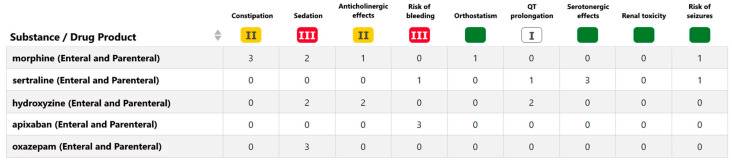
Overview of Janusmed Risk Profile for a hypothetical patient. On the left, the list of drug products (substances) that the patient is taking. For each substance, a risk value is shown for each of the nine risk categories. For each risk category, the risk values for all substances are used to calculate a risk level as follows: no increased risk (green), risk level I (somewhat increased risk, white), risk level II (moderate increased risk, yellow), and risk level III (significant increased risk, red).

**Figure 2 pharmacy-12-00168-f002:**
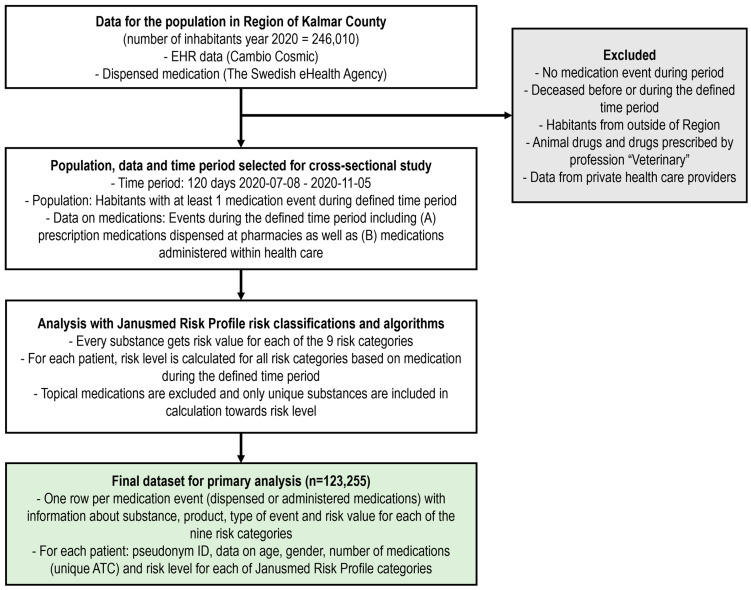
Flowchart describing the data used in this study. The defined 120-day period was used for all analysis to capture simultaneous use of medications. Patients with no medication events in our data during the period were excluded, as well as patients who died before or during the defined period, or patients that were not habitants of Kalmar. In addition, data on veterinary medications were excluded, i.e., those with an ATC code that started with Q or where the prescriber was a veterinarian. We also did not use data from privately owned health care providers. EHR = electronic health record, ATC = the Anatomical Therapeutic Chemical (ATC) classification system where the 5th level was used.

**Figure 3 pharmacy-12-00168-f003:**
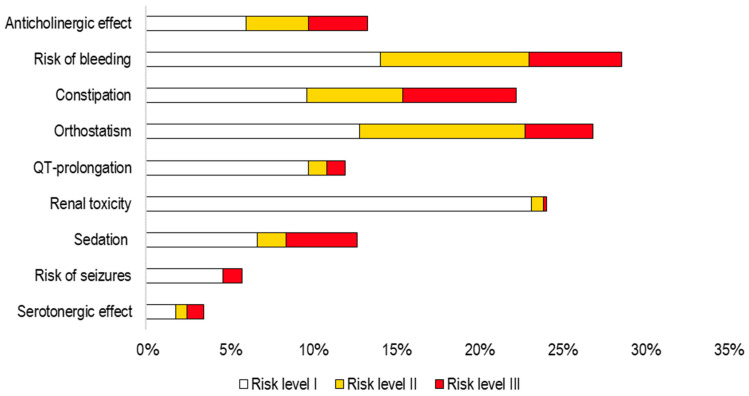
Proportion of patients (%) with a potential risk for each of the nine risk categories according to Janusmed Risk Profile algorithms, divided into the three risk levels: risk level I (somewhat increased risk, white), risk level II (moderate increased risk, yellow), and risk level III (significant increased risk, red). Calculated with a combined measure of dispensed prescriptions and administered at hospital during the defined 120-day period. Patients with at least one medication in that time period in Kalmar (n = 123,255) are included.

**Figure 4 pharmacy-12-00168-f004:**
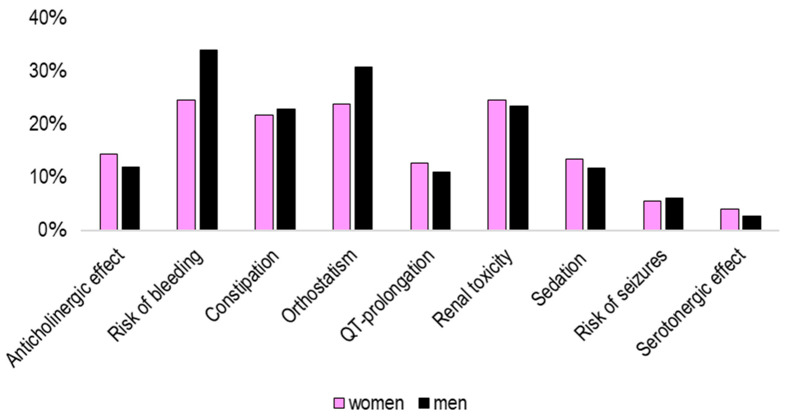
Proportion of patients with a potential risk for each of the nine risk categories according to Janusmed Risk Profile algorithms; comparison between females and males (%). All three risk levels (I, II, and III) are included in the proportion. Calculated with a combined measure of dispensed prescriptions and administered at hospital in the defined 120-day period. Proportion among patients with at least 1 medication (n = 123,255). The difference was statistically significant for each category (Mann–Whitney U-test, *p* ≤ 0.001).

**Figure 5 pharmacy-12-00168-f005:**
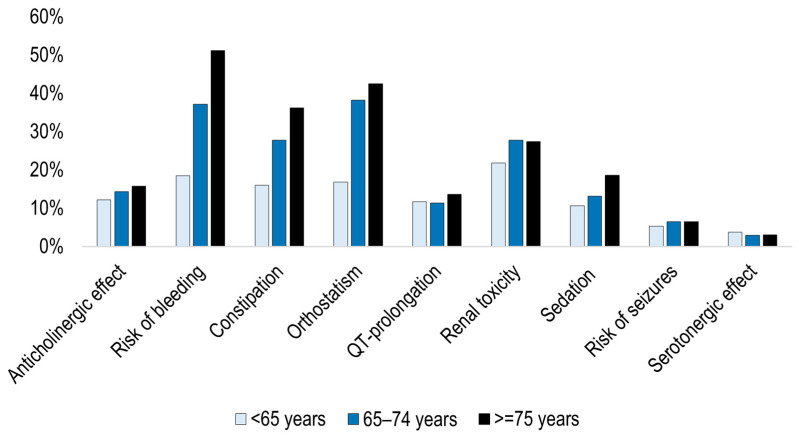
Proportion of patients with a potential risk for each of the nine risk categories according to Janusmed Risk Profile algorithms; comparison between age groups (%). All three risk levels (I, II, and III) are included in the proportion. Calculated with a combined measure of dispensed prescriptions and administered at hospital in the defined 120-day period. Proportion among patients with at least 1 medication (n = 123,255). The difference was statistically significant for each category (Kruskal–Wallis test, *p* < 0.001).

**Figure 6 pharmacy-12-00168-f006:**
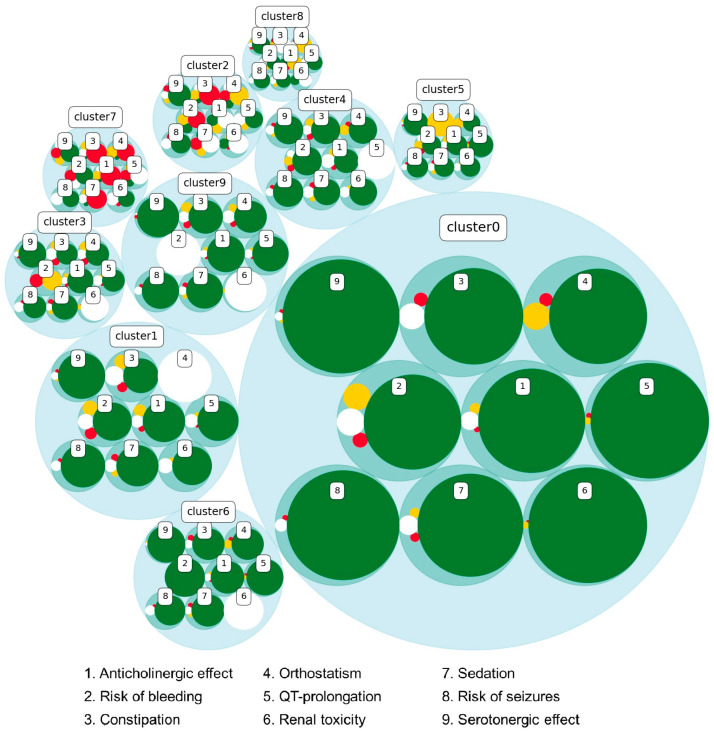
Cluster analysis of the risk profile for all individuals in the population with at least one medication in the defined 120-day period in Kalmar (n = 123,255). Ten clusters were identified based on the risk profile of each individual, putting patients with similar profiles in the same cluster. The size of the circle for the cluster is decided by the number of individuals in that cluster, and, in the cluster circle, the proportion of patients with an increased risk of each of the nine categories is visualized as no increased risk (green), risk level I (somewhat increased risk, white), risk level II (moderate increased risk, yellow), and risk level III (significant increased risk, red). Descriptions of the patients in each cluster are provided in Table 2 and Appendix B. The nine pharmacological risk categories are numbered as (1) anticholinergic effect, (2) risk of bleeding, (3) constipation, (4) orthostatism, (5) QT prolongation, (6) renal toxicity, (7) sedation, (8) risk of seizures, and (9) serotonergic effect.

**Table 1 pharmacy-12-00168-t001:** Top 10 medications (summarized risk value) within each risk category among patients with risk level III or higher. Substance (ATC) [risk value]. Summarized risk value was calculated for each risk category for patients with risk level III for that category as number of patients with the substance multiplied by the risk value for substance.

Anticholinergic Effect	Risk of Bleeding	Constipation	Orthostatism	QT Prolongation	Renal Toxicity	Sedation	Risk of Seizures	Serotonergic Effect
Amitriptyline (N06AA09) [3]	Acetylsalicylic acid, low-dose (B01AC06) [2]	Oxycodone (N02AA05) [3]	Losartan (C09CA01) [2]	Ondansetron (A04AA01) [2]	Omeprazole (A02BC01) [1]	Zopiclone (N05CF01) [2]	Levomepromazine (N05AA02) [3]	Sertraline (N06AB06) [3]
Promethazine (R06AD02) [3]	Apixaban (B01AF02) [3]	Morphine (N02AA01) [3]	Oxycodone (N02AA05) [2]	Ciprofloxacin (J01MA02) [2]	Naproxen (M01AE02) [1]	Oxazepam (N05BA04) [3]	Ciclosporin (L04AD01) [3]	Venlafaxine (N06AX16) [3]
Hydroxyzine (N05BB01) [2]	Tinzaparin (B01AB10) [3]	Atorvastatin (C10AA05) [1]	Isosorbide mononitrate (C01DA14) [3]	Oxycodone (N02AA05) [1]	Esomeprazole (A02BC05) [1]	Oxycodone (N02AA05) [2]	Clozapine (N05AH02) [3]	Tramadol (N02AX02) [3]
Oxycodone (N02AA05) [1]	Clopidogrel (B01AC04) [3]	Paracetamol, codeine (N02AJ06) [2]	Glyceryl trinitrate (C01DA02) [3]	Hydroxyzine (N05BB01) [2]	Diclofenac (M01AB05) [1]	Diazepam (N05BA01) [3]	Olanzapine (N05AH03) [2]	Duloxetine (N06AX21) [3]
Tiotropium (R03BB04) [2]	Naproxen (M01AE02) [2]	Loperamide (A07DA03) [3]	Amitriptyline (N06AA09) [3]	Escitalopram (N06AB10) [2]	Ibuprofen (M01AE01) [1]	Mirtazapine (N06AX11) [2]	Quetiapine (N05AH04) [2]	Escitalopram (N06AB10) [3]
Olanzapine (N05AH03) [3]	Warfarin (B01AA03) [3]	Iron (B03AA07) [3]	Carvedilol (C07AG02) [3]	Mirtazapine (N06AX11) [1]	Methotrexate (L04AX03) [1]	Morphine (N02AA01) [2]	Alimemazine (R06AD01) [1]	Citalopram (N06AB04) [3]
Alimemazine (R06AD01) [2]	Ibuprofen (M01AE01) [2]	Furosemide (C03CA01) [1]	Losartan/hydrochlorothiazide (C09DA01) [2/1]	Sertraline (N06AB06) [1]	Ketoprofen (M01AE03) [1]	Propiomazine (N05CM06) [2]	Propiomazine (N05CM06) [1]	Amitriptyline (N06AA09) [2]
Mirtazapine (N06AX11) [1]	Diclofenac (M01AB05) [2]	Ketobemidone (N02AB01) [3]	Enalapril (C09AA02) [1]	Citalopram (N06AB04) [2]	Etoricoxib (M01AH05) [1]	Zolpidem (N05CF02) [2]	Oxycodone (N02AA05) [1]	Metoclopramide (A03FA01) [2]
Propiomazine (N05CM06) [1]	Prednisolone (H02AB06) [1]	Amitriptyline (N06AA09) [2]	Promethazine (R06AD02) [2]	Metoclopramide (A03FA01) [1]	Valaciclovir (J05AB11) [1]	Midazolam (N05CD08) [3]	Zuclopenthixol (N05AF05) [2]	Oxycodone (N02AA05) [1]
Quetiapine (N05AH04) [2]	Ticagrelor (B01AC24) [3]	Bisoprolol (C07AB07) [1]	Doxazosin (C02CA04) [3]	Fluconazole (J02AC01) [2]	Sulfasalazine (A07EC01) [1]	Gabapentin (N03AX12) [2]	Mycophenolic acid (L04AA06) [1]	Fluoxetine (N06AB03) [3]

**Table 2 pharmacy-12-00168-t002:** Overview of characteristics of patients in each of the 10 clusters. Based on gender and age, number of patients and proportion within that cluster are described. Number of medications is calculated as unique substances (ATC 5th level) during the defined 120-day period. Risk profile sum is calculated for each patient as the summary of the risk level for each of the 9 risk categories, i.e., could range between 0 and 27. Number of medications and risk profile sum are presented as median and interquartile range (IQR).

Cluster	Female, n (%)	Male, n (%)	Age < 65, n (%)	Age 65–74, n (%)	Age ≥ 75, n (%)	Number of Medications (Median; IQR)	Risk Profile Sum (Median; IQR)
0	39,990 (56.5)	30,795 (43.5)	46,728 (66.0)	11,131 (15.7)	12,926 (18.3)	2; 3	0; 1
1	6424 (48.9)	6707 (51.1)	5477 (41.7)	3389 (25.8)	4265 (32.5)	4; 4	2; 3
2	2176 (56.6)	1670 (43.4)	1493 (38.8)	848 (22.0)	1505 (39.1)	11; 7	11; 5
3	2414 (53.0)	2143 (47.0)	1874 (41.1)	990 (21.7)	1693 (37.2)	7; 5	5; 2
4	3859 (61.6)	2409 (38.4)	4185 (66.8)	890 (14.2)	1193 (19.1)	4; 4	2; 3
5	1816 (58.0)	1317 (42.0)	1523 (48.6)	606 (19.3)	1004 (32.0)	5; 5	4; 3
6	4166 (58.6)	2945 (41.4)	4177 (58.7)	1534 (21.6)	1400 (19.7)	3; 3	1; 1
7	2037 (57.7)	1494 (42.3)	1925 (54.5)	619 (17.5)	987 (28.0)	11; 10	15; 6
8	1320 (65.3)	702 (34.7)	1154 (57.1)	428 (21.2)	440 (21.8)	6; 5	6; 3
9	4908 (55.3)	3963 (44.7)	5965 (67.2)	1465 (16.5)	1441 (16.2)	4; 4	2; 1

## Data Availability

Data are not made available due to legal and ethical restrictions. Data were obtained from a third party. For questions, please contact the corresponding author.

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
