# Peer review of "Potential Adverse Drug Events Identified with Decision Support Algorithms from Janusmed Risk Profile—A Retrospective Population-Based Study in a Swedish Region"

_pharmacy, 2024, doi:10.3390/pharmacy12060168_

Round 1
Reviewer 1 Report
Comments and Suggestions for Authors
Dear Authors,
This is an interesting research. Also useful to bridge the usual gap of "prevention" in pharmacovigilance activities. We are quite good at detecting adverse reactions but less good at avoiding them through a more careful selection, prescription, dispensing and use.
Janusmed Risk Profile is a potentially useful approach. And this retrospective study provides interesting findings.
However, in my opinion, the study would greatly improve their interest by trying to identify adverse drug events observed in the included patients, both those who showed a risk profile according to the JRP and those who did not. I would like to see the type of adverse event, the medicines involved, etc.
Author Response
Dear reviewer
Thank you very much for taking the time to review our manuscript. We are grateful for your feedback and have revised the manuscript accordingly where possible. Please find the detailed responses below and the corresponding revisions in track changes in the re-submitted manuscript file. In the attached file you can see responses to all four reviewers and the editor.
Reviewer 1, Comment 1: This is an interesting research. Also useful to bridge the usual gap of "prevention" in pharmacovigilance activities. We are quite good at detecting adverse reactions but less good at avoiding them through a more careful selection, prescription, dispensing and use.
Response: Thank you! We hope that this study and the following studies we are doing in our research group can contribute to knowledge on that topic.
Reviewer 1, Comment 2: Janusmed Risk Profile is a potentially useful approach. And this retrospective study provides interesting findings. However, in my opinion, the study would greatly improve their interest by trying to identify adverse drug events observed in the included patients, both those who showed a risk profile according to the JRP and those who did not. I would like to see the type of adverse event, the medicines involved, etc.
Response: We are glad that you mention this. In our ongoing research we are doing exactly that. This first paper only aims at describing the potential ADEs so that we have an overview for the continued research. In the last section of discussion in the paper we described this and added a little more to clarify this as a response to your question (line 450-455).

Reviewer 2 Report
Comments and Suggestions for Authors
Thanks for the opportunity to revise this nice article dealing with that one important aspect of prescription which involved for both clinical and pharmaceutical professionals.
The article is well written and the methods are clearly stated at the beginning of the paper. Maybe the introduction I could be shortened a little bit.
There are some minor revision to do on the reference list and the text as in :
line 425 : Based on previous research (ref?) it is .. Reference e is missing
One of the most important are missing a point to ease the absence of having detected the true adverse events are in this cohort. Moreover one of the pointed out could be discussed the father in the discussion part is the fact that they associated clinical relevance to age and gender distribution in the augmentation of risk, as also the authors stated this kind of tools should also examine the clinical relevance of the CDSS classifications. It is pretty common that younger patients with chronic condition have sometimes much more daily medication than older people. But these informations regarding the clinical context of the cohort is not detailed. This point should also be listed in the limitation part and one other point of discussion should be how to integrate easily some relevant clinical condition to this kind of scores. There are many other scores indicating and trying to evaluating the complexity of a pharmaceutical regimen based for example on the number of medication and their characteristics. Is there an integration of multiple scores in this field that can be applied in clinical practice ?
Author Response
Dear reviewer
Thank you very much for taking the time to review our manuscript. We are grateful for your feedback and have revised the manuscript accordingly where possible. Please find the detailed responses below and the corresponding revisions in track changes in the re-submitted manuscript file. In the attached file you can see responses to all four reviewers and the editor.
Reviewer 2, Comment 1: Thanks for the opportunity to revise this nice article dealing with that one important aspect of prescription which involved for both clinical and pharmaceutical professionals.
Response: Thank you, we are glad you appreciate the paper.
Reviewer 2, Comment 2: The article is well written and the methods are clearly stated at the beginning of the paper. Maybe the introduction I could be shortened a little bit.
Response: We agree that the introduction is a little long. We have removed two sentences but feel that the rest is needed to both understand the background, and Janusmed Risk Profile.
Reviewer 2, Comment 3: There are some minor revision to do on the reference list and the text as in :
line 425 : Based on previous research (ref?) it is .. Reference e is missing
Response: Thank you for finding this error. We have now added references there.
Reviewer 2, Comment 4: One of the most important are missing a point to ease the absence of having detected the true adverse events are in this cohort. Moreover one of the pointed out could be discussed the father in the discussion part is the fact that they associated clinical relevance to age and gender distribution in the augmentation of risk, as also the authors stated this kind of tools should also examine the clinical relevance of the CDSS classifications. It is pretty common that younger patients with chronic condition have sometimes much more daily medication than older people. But these information regarding the clinical context of the cohort is not detailed. This point should also be listed in the limitation part and one other point of discussion should be how to integrate easily some relevant clinical condition to this kind of scores. There are many other scores indicating and trying to evaluate the complexity of a pharmaceutical regimen based for example on the number of medication and their characteristics. Is there an integration of multiple scores in this field that can be applied in clinical practice?
Response: Thank you for this feedback. There are many different ways of scoring risks for ADEs and it is important that each method is validated in real world data. In the limitation part we have added weaknesses in that we are not including other data such as patient characteristics (line 442-446). We have also clarified that we are continuing with other studies to examine clinical relevance of alerts (line 451-455).

Reviewer 3 Report
Comments and Suggestions for Authors
High quality paper describing how one specific decision support algorithm can be used to quantify possible adverse effects in a large population.
The authors describe well the advantages and limitations of the retrospective study, and of the used tool. I would have hoped to see the aspects of drug dosage and patient characteristics mentionned (age, weight, co-morbities, etc.) as contributing or mitigating factors when calculating cumulative risk.
Some detailled remarks
- Line 65 : Janusmed Risk Profile is said to be a “unique” CDSS. It is not clear in what sense in is unique. There are other systems out there that assess cumulative risk.
- Figure 4 and 5 captions : specify that the shown risks concern all levels (I, II, and III)
- Line 425 : please add correct reference for “previous research”
- Appendix A : correct spelling is documented, and not documentet
Author Response
Dear reviewer
Thank you very much for taking the time to review our manuscript. We are grateful for your feedback and have revised the manuscript accordingly where possible. Please find the detailed responses below and the corresponding revisions in track changes in the re-submitted manuscript file. In the attached file you can see responses to all four reviewers and the editor.
---
Reviewer 3, Comment 1: High quality paper describing how one specific decision support algorithm can be used to quantify possible adverse effects in a large population. The authors describe well the advantages and limitations of the retrospective study, and of the used tool. I would have hoped to see the aspects of drug dosage and patient characteristics mentionned (age, weight, co-morbities, etc.) as contributing or mitigating factors when calculating cumulative risk.
Response: Thank you! Interesting point. Drug dosage and patient characteristics mentioned (age, weight, co-morbities, etc.) would indeed be valuable, but is outside the scope of this specific paper. However, we are looking into those aspect in our continued research. We have added a couple of sentences about weaknesses and future research (line 442-446 and line 451-455) as a response to this.
Reviewer 3, Comment 2: Line 65 : Janusmed Risk Profile is said to be a “unique” CDSS. It is not clear in what sense in is unique. There are other systems out there that assess cumulative risk.
Response: We have discussed this among the authors and decided to remove the word unique from the manuscript, as it is perhaps a too strong word. As far as we know there are no identical systems, but there are systems with similar functionality at least for some of the risk categories in Janusmed Risk Profile.
Reviewer 3, Comment 3: Figure 4 and 5 captions : specify that the shown risks concern all levels (I, II, and III)
Response: Good point! We have added this information for figure 4 and 5 to make it clear.
Reviewer 3, Comment 4: Line 425 : please add correct reference for “previous research”
Response: Thank you for noticing this error. We have added the actual references now.
Reviewer 3, Comment 5: Appendix A : correct spelling is documented, and not documentet
Response: Once again, thank you for being so thorough and seeing this error. We have corrected and updated with a new version of the figure in Appendix A.

Reviewer 4 Report
Comments and Suggestions for Authors
Manuscript ID: pharmacy-3266693
This is an interesting study on the prevalence of potential ADEs identified using the CDSS algorithms from the Janusmed risk profile covering the population of a Swedish region. A detailed sample size estimate and using CDSS algorithms from the Janusmed Risk Profile provide a solid framework for analysis. However, there are a few areas where it could be improved:
Abstract section. Material and Methods, and Conclusion require more data to give readers a better insight into this retrospective study. The Results should be more specific (e.g. which “combinations of medications increase the risk of ADEs”). It is necessary for the authors to explain why they used the term adverse drug events in the first part of the abstract, while in the second part they used the term side effects. This also applies to the rest of the manuscript.
Keywords: Is it really necessary to list words that are not mentioned at all in the abstract? This gives the impression that the authors have no insight into the research itself. The abstract is a separate entity from the rest of the manuscript. It is necessary that all keywords that are contained in the abstract and the rest of the text are listed in keywords.
Introduction section. It is not recommended to include such sentences in the Introduction: “In this we include adverse drug reactions (ADRs) and pharmacokinetic drug-drug interactions (DDIs) and pharmacodynamic DDIs among other things.”
Results section. It is not necessary to repeat the names of the statistical tests that were used, they are already listed in the Materials and Methods section.
Line 290-292. This sentence requires correction to make it more understandable for readers.
Discussion section.
Line 376-389. The authors only state the results of their research and compare them with previous findings, but in no segment do they provide possible explanations for the obtained results.
References section.
Technical corrections of incomplete references are necessary in the References section (ref. 11, 27).

Author Response
Dear reviewer
Thank you very much for taking the time to review our manuscript. We are grateful for your feedback and have revised the manuscript accordingly where possible. Please find the detailed responses below and the corresponding revisions in track changes in the re-submitted manuscript file. In the attached file you can see responses to all four reviewers and the editor.
---
Reviewer 4, Comment 1: This is an interesting study on the prevalence of potential ADEs identified using the CDSS algorithms from the Janusmed risk profile covering the population of a Swedish region. A detailed sample size estimate and using CDSS algorithms from the Janusmed Risk Profile provide a solid framework for analysis. However, there are a few areas where it could be improved.
Response: Thank you for this and taking time to review and provide valuable feedback.
Reviewer 4, Comment 2: Abstract section. Material and Methods, and Conclusion require more data to give readers a better insight into this retrospective study. The Results should be more specific (e.g. which “combinations of medications increase the risk of ADEs”). It is necessary for the authors to explain why they used the term adverse drug events in the first part of the abstract, while in the second part they used the term side effects. This also applies to the rest of the manuscript.
Response: We have revised the abstract and made several changes to hopefully make it better. However, as the word limit is 200 words it is difficult to be very specific. Thank you for seeing the inconsistencies in how we used ADE and side effects. We have changed this to ADEs more consistently in the abstract and also checked the manuscript to see that we are using ADEs and side effects in a consistent way. Although we believe that ADE (or potential ADE) is a more scientifically correct term in this paper we still use side effects in some places as it is how Janusmed Risk profile describe their risk categories.
Reviewer 4, Comment 3: Keywords: Is it really necessary to list words that are not mentioned at all in the abstract? This gives the impression that the authors have no insight into the research itself. The abstract is a separate entity from the rest of the manuscript. It is necessary that all keywords that are contained in the abstract and the rest of the text are listed in keywords.
Response: We have removed two keywords and added one. Sometimes different words are used to describe a similar topic, therefore we wanted to include keywords that would capture those alternative words that are not included in the abstract or manuscript. For example, this study could be interesting for researchers working with potential drug-related problems.
Reviewer 4, Comment 4: Introduction section. It is not recommended to include such sentences in the Introduction: “In this we include adverse drug reactions (ADRs) and pharmacokinetic drug-drug interactions (DDIs) and pharmacodynamic DDIs among other things.”
Response: We have changed this sentence.
Reviewer 4, Comment 5: Results section. It is not necessary to repeat the names of the statistical tests that were used, they are already listed in the Materials and Methods section.
Response: We have removed the name of the statistical tests from the results section.
Reviewer 4, Comment 6: Line 290-292. This sentence requires correction to make it more understandable for readers.
Response: Thank you for pointing this out. We have re-written parts of this section, hopefully making it clearer.
Reviewer 4, Comment 7: Discussion section. Line 376-389. The authors only state the results of their research and compare them with previous findings, but in no segment do they provide possible explanations for the obtained results.
Response: Good point! We have revised this section to clarify, added some information and possible explanation.
Reviewer 4, Comment 8: References section. Technical corrections of incomplete references are necessary in the References section (ref. 11, 27).
Response: Thank you for noticing this. We have corrected those references. (note that the references have now changed number as we added a number of new references)
